# Corrosion Inhibition Mechanism of Ultra-High-Temperature Acidizing Corrosion Inhibitor for 2205 Duplex Stainless Steel

**DOI:** 10.3390/ma16062358

**Published:** 2023-03-15

**Authors:** Danping Li, Wenwen Song, Junping Zhang, Chengxian Yin, Mifeng Zhao, Hongzhou Chao, Juantao Zhang, Zigang Lei, Lei Fan, Wan Liu, Xiaolong Li

**Affiliations:** 1Tubular Goods Research Institute of CNPC, Xi’an 710077, China; 2Xi’an Dexincheng Technology Co., Ltd., Xi’an 710075, China; 3PetroChina Tarim Oilfield Company, Korla 841000, China; 4School of Chemistry and Chemical Engineering, Northwestern Polytechnical University, Xi’an 710082, China

**Keywords:** duplex stainless steel, selective corrosion, high-temperature acidification, corrosion inhibitor, molecular dynamics simulation

## Abstract

The acidizing corrosion inhibitors reported so far have a poor effect on duplex stainless steel in high-temperature and high-concentration acid systems and cannot effectively inhibit the occurrence of selective corrosion. In this paper, a new acidizing corrosion inhibitor was designed, which was mainly composed of Mannich base and antimony salt. The inorganic substance in the corrosion inhibitor had good stability at high temperatures and could quickly form a complex with the metal matrix to enhance the binding ability. The organic substance can make up for the non-dense part of the inorganic film. The properties of developed corrosion inhibitors were analyzed by quantum chemical calculation, molecular dynamics simulation, and scanning electron microscopy. The results showed that a double-layer membrane structure could be constructed after adding the corrosion inhibitor, which could play a good role in blocking the diffusion of acid solution at high-temperature. The uniform corrosion rate of 2205 duplex stainless steel after adding acidizing corrosion inhibitor immersion in a simulated service condition (9 wt.% HCl + 1.5 wt.% HF + 3 wt.% CH_3_COOH + 4~6 wt.%) at 140 °C, 160 °C and 180 °C for a 4 h test is 6.9350 g·m^−2^·h^−1^, 6.3899 g·m^−2^·h^−1^ and 12.1881 g·m^−2^·h^−1^, respectively, which shows excellent corrosion inhibition effect and is far lower than that of the commonly accepted 81 g·m^−2^·h^−1^ and no selective corrosion could be detected.

## 1. Introduction

With the increasing demand for energy, the exploration and development of oil and gas fields have gradually developed from conventional working conditions to the harsh working conditions of high-temperature and high-pressure or even ultra-high-temperature and high-pressure. The high-temperature and high-pressure well is defined with wellhead pressure and bottom hole pressure greater than 70 and 105 MPa, respectively, and the bottom hole temperature greater than 150 °C by the International high-temperature and high-pressure Well Association, the wells with wellhead pressure greater than 105 MPa, bottom-hole pressure greater than 140 MPa and bottom-hole temperature greater than 175 °C is defined as ultra-high-temperature and pressure wells [1]. At present, high-temperature and high-pressure oil and gas wells in the world are mainly distributed in the Gulf of Mexico in the United States, the North Sea in the United Kingdom, Southeast Asia, Africa, and the Tarim Basin, the South China Sea and Sichuan in China [2,3,4,5]. The acidizing operation temperature of oil wells in the North Sea has exceeded 160 °C [6]. In China, representative ultra-high-temperature and high-pressure wells are mainly distributed in the Tarim Basin of Xinjiang, with deep burial depths and extremely harsh service conditions. The bottom hole temperature usually exceeds 180 °C [7,8]. In addition, the acid with extremely low pH value in the process of reservoir reconstruction and the CO_2_ and Cl^−^ in the fluid produced in the later production process all put forward high requirements for the corrosion resistance of oil pipes. According to the SUMITOMO oil well pipe material selection spectrum [9], duplex stainless steel should be used for H_2_S partial pressure ≤0.1 bar when the temperature is over 180 °C, while nickel base alloy with higher corrosion resistance should be selected for H_2_S partial pressure > 0.1 bar, and the traditional super 13Cr stainless steel could no longer meet the requirement of the working conditions.

Generally, a passive oxide (mainly Cr_2_O_3_) layer could be formed on the stainless steel to resist various corrosion, and the absence or breakdown of this layer could result in accelerated corrosion rates [10]. Stainless steel is mainly divided into the following categories: ferritic stainless steels, austenitic stainless steels, martensitic stainless steels, duplex stainless steels, precipitation-hardening stainless steels, and Mn-N substituted austenitic stainless steels [11]. Due to the particularity of its composition, duplex stainless steel has the characteristics of ferrite and austenite stainless steel. Duplex stainless steel has higher plastic toughness, and its weldability and intergranular corrosion resistance are significantly stronger than that of ferritic stainless steel. Compared with austenitic stainless steel, the yield strength, intergranular corrosion, and Cl^−^ corrosion resistance of duplex steel is much better. In addition, duplex stainless steel also has the advantage of better resistance to stress corrosion, pitting, and crevice corrosion [12].

During the whole production cycle of the well, applying a strong acid (such as 10–28 wt.% HCl) to increase the output of the reservoir is inevitable. The passive film of duplex stainless steel could be destroyed under such harsh conditions. Duplex stainless steel has good corrosion resistance in general conditions, but it will exhibit selective corrosion of different phases in specific conditions [13,14,15,16,17,18,19,20,21,22,23,24,25]. Due to the different crystal structure and chemical composition of α and γ phases, which shows different electrochemical potential in a specific solution, a micro-electric couple will be formed and cause selective corrosion [26,27]. For example, the anodic polarization curve of 2205 duplex stainless steel sample in acid solution usually shows two anodic dissolution peaks, the lower and the higher peaks are the ferrite phase and austenite phase activation peaks, respectively. The ferrite phase is preferentially dissolved when the applied potential corresponds to the lower peak. Similarly, the austenite phase is preferentially dissolved when the applied potential is close to the higher peak [16]. Yau and Streicher [24] found that ferritic selective corrosion of FeCr–10%Ni duplex stainless steel occurred in reducing acid. Sridhar and Kolts [25] discovered that the austenitic phase selective corrosion of high nitrogen content duplex stainless steel occurred in sulfuric acid and phosphoric acid, while the ferrite phase selective corrosion occurred in hydrochloric acid. Tsai et al. [18] showed that the activity of ferrite was stronger than austenite in 2 mol/L H_2_SO_4_ + 0.5 mol/L HCl solution; however, austenite was in a more active state and is preferentially corroded as the anode in 1.5 mol/L HNO_3_ solution.

Acid corrosion inhibitors must be used to reduce the corrosion damage of steel in service [26]. At present, most of the research on acidizing corrosion inhibitors for duplex stainless steel was mainly focused on the acid solution system under 120 °C [12,27,28,29,30] or high-temperature, low-concentration acid [31]. Hirotaka et al. [28] adopted 7 wt.% HCl at 60 °C as the test medium to simulate the acidizing bottom-hole conditions of onshore oil fields in Japan. The corrosion inhibition mechanism of 25 wt.% cinnamaldehyde + 20 wt.% long-chain alkyl imidazoline + 55 wt.% methanol plus CuI/KI/Cu/CuCl_2_/CuSO_4_ on 2205 duplex stainless steel in an acid system was studied, the results showed that the corrosion inhibition rate was only 34.9 wt.%, and the ferrite selective corrosion could be observed, and the corrosion inhibition rate reaches more than 98% after the addition of Cu compound, and no selective corrosion could be seen. Wang et al. [12] showed that the corrosion rate of 2205 duplex stainless steel was 26.8053 g·m^−2^·h^−1^ after adding corrosion inhibitor in 10 wt.% HCl + 1.5% wt.HF + 3 wt.% HAC solution at 120 °C, but ferrite selective corrosion could be seen on the surface of the sample. Du et al. [31] showed that the corrosion rate of 2205 duplex stainless steel could be controlled at 40.3092 g·m^−2^·h^−1^ in the 4 wt.% HCl + 3 wt.% CH_3_COOH + 5 wt.% QASE + 1 wt.% Sb_2_O_3_ at 200 °C test environment.

Molecular simulation techniques play an important role in the modern characterization of materials [32], which are beneficial to the study of corrosion behavior by increasing the understanding of chemical and physical processes at the molecular and atomic levels. Cl^−^ has a great influence on metal pitting corrosion, and the molecular dynamics (MD) simulations method is also used to study the pitting behavior of Cl^−^ on metals [32,33,34,35,36]. By using molecular dynamics simulation, Sepehr Y et al. [33] studied the pitting corrosion behavior of nano-diamonds. Chen et al. [34], using MD simulation, found that the mixed corrosive anions (Cl^−^ and HSO_3_^−^) adsorbed much more strongly to the passive film, and their diffusion coefficient was also significantly improved compared to the solution containing only one kind of anion.

The above-mentioned acidizing corrosion inhibitor had a certain corrosion inhibition effect on 2205 duplex stainless steel. However, there have been no reports on corrosion inhibitors suitable for ultra-high temperature and high-concentration acid solution systems. Herein, it is essential to develop a corrosion inhibitor suitable for the ultra-high-temperature and high-concentration acidizing environment and to clarify its corrosion inhibition mechanism, which is critical to prolonging the service life of duplex stainless steel. In this paper, quantum chemical calculation and molecular dynamics simulation were applied to design a new acid corrosion inhibitor, and the suitability was further verified by high-temperature and high-pressure corrosion simulation tests. This research has great engineering application value to broaden the selection of materials for ultra-high-temperature and high-pressure wells and reduce the cost of oil and gas field string.

## 2. Research Methods

### 2.1. Test Materials

The special ultra-high-temperature acidizing corrosion inhibitor for duplex stainless steel was developed by China Petroleum Engineering Materials Research Institute Co., Ltd., Xi‘an, China. and it is mainly composed of 40 wt.% Mannich base + 6 wt.% antimony trioxide + 15 wt.% hydrochloric acid + 5 wt.% OP + 34 wt.% DMF.

The test 2205 duplex stainless steel was developed and produced by Zhejiang Jiuli Special Material Technology Co., Ltd., Huzhou China. and its chemical composition was 0.019 wt.% C, 0.41 wt.%Si, 0.63 wt.% Mn, 0.023 wt.% P, 0.005 wt.% S, 2.38 wt.% Cr, 52 wt.% Mo, 5.47 wt.% Ni, 0.181 wt.%N and residual Fe. The size of the high-temperature and high-pressure corrosion test sample was cut from a pipe material with the dimension of 50 mm × 10 mm × 3 mm.

Before testing, the samples were ground by using the abrasive paper from 400 # to 800 # step by step, then immersed in anhydrous ethanol for 5 min by ultrasonic cleaning, dried with cold air, placed in a dryer for at least 1 h, weighed (accurate to 0.0001 g), and measured the size (accurate to 0.01 mm). The samples should be mutually insulated and installed on the fixture during the test. After testing, the acid solution on the surface of the sample was removed by deionization, the corrosion inhibitor film on the surface of the sample was removed by N, N–dimethylformamide, and then the sample was ultrasonically cleaned with anhydrous ethanol for 5 min, dried by cold air, placed in a desiccator for at least 1 h and weighed (accurate to 0.0001 g).

### 2.2. High-Temperature and High-Pressure Corrosion Test

#### 2.2.1. Test Conditions

The typical operating temperature and acidizing operation system of an oil and gas field in western China were selected to evaluate the performance of corrosion inhibitors. The test period was selected based on normal operation time and possible long operation time in the oil field. Table 1 shows the specific test conditions.

The high-temperature and high-pressure test was carried out in TFCZ5 25/450 autoclave, which the test temperature ≤ 450 °C and pressure ≤ 25 MPa.

The uniform corrosion rate was calculated by the weight-loss method according to the following equation:(1)$$V=\frac{10^{6}\Delta m}{A\cdot \Delta t}$$
where *V* is uniform corrosion rate; g·m^−2^.h^−1^; Δ*t* is reaction time, h; Δ*m* is weight loss, g; and *A* is surface area of test sample, mm^2^.

#### 2.2.2. Microstructure Analysis

The depth of local corrosion was measured and counted by the Smart Zoom5 (Smart Zoom 5, Zeiss, Oberkochen, Germany), with a magnification of 10–500×. The phase composition of 2205 duplex stainless steel was analyzed by an X-ray diffractometer (SmartLab, Rigaku, Tokyo, Japan). The step length was 0.02°, the time of each step was 5 s, and the angular spacing was between 30° and 90°. Scanning electron microscope (SEM, JSM-IT500LA, JEOL Ltd., Japan) and energy spectrum analysis (EDS, JEOL Ltd., Japan) were used to test the corrosion morphology, inhibitor film thickness, and the composition of the film after high-temperature and high-pressure tests. The magnification was 5–300,000×, the secondary electron was 3.0 nm (30 kV), and the backscattered electron was 4.0 mm (30 kV). When measuring the thickness of the film, the sample was embedded in the epoxy resin, and a 50 mm × 3 mm section was observed. Before observing the inner film of the corrosion inhibitor, soak the sample in N, N-dimethylformamide for 10 s to remove the outer film, and then wash it with clean water, dehydrate it with anhydrous ethanol, and dry it with cold air.

### 2.3. Molecular Dynamics Simulation

The molecular dynamics simulation software Accelrys MS Modeling 7.2 (Accelrys, San Diego, CA, USA) was used to build the model of the water-Fe corrosion inhibitor system, and the diffusion behavior of the corrosion medium particles H_2_O, H_3_O^+^, and Cl^−^ in the corrosion inhibitor film was established and calculated.

#### 2.3.1. Construction of Water-Fe-Corrosion Inhibitor System Model

Firstly, construct a cell of metal Fe, cut the cell along the (001) plane into a surface with a thickness of 17.198 Å, and construct it into (11 × 11) Two-dimensional supercell, and then build it as 31.53 Å × 31.53 Å × 15.76 Å three-dimensional supercell, and 31.53 Å × 31.53 Å × 24.06 Å of liquid water three-dimensional amorphous unit is constructed by the Amorphous Cell module. The constructed inhibitor molecules were immersed into the amorphous unit. The molecular positions of water and corrosion inhibitor are determined randomly. Finally, a three-layer supercell structure was constructed by the “vacuum layer-corrosion inhibitor solution layer-metal Fe supercell” in top-down order. In order to study the interaction between the inhibitor molecule and the Fe surface, the inhibitor molecule was manually moved to the appropriate position near the Fe surface, and the position was appropriate to not form a short-range interaction. The simulation process adopts a group-based truncation method (that is, group-based truncation), with a truncation radius of 9.5 Å, to ensure the calculation accuracy and minimize the calculation time. Then the Fe layer of the final structure is fixed. The smart method is used to minimize the energy of the built three-layer supercell structure by 5000 steps to remove the local high potential energy points for dynamic balance and data acquisition.

The NVT ensemble is used for dynamic balance and data acquisition. The Andersen temperature control method was used to conduct dynamic balance of 160 ps at first and then conduct data acquisition of 80 ps. One track file was output every 800 fs, and a total of 100 track files were output. The time step in the simulation process was 0.8 fs. COMPASS force field was used in the simulation process.

#### 2.3.2. Establishment and Calculation of Diffusion Behavior Model of Corrosion Medium Particles H_2_O and H_3_O^+^ in Corrosion Inhibitor Film

Three corrosion particles, H_2_O, H_3_O^+^, and Cl^−^, were selected. The simulation system consists of one corrosion medium particle and 100 corrosion inhibitor molecules. Firstly, the Amorphous Cell module was used to build an amorphous structure containing 100 corrosion inhibitor molecules with periodic boundary conditions, and then the NPT ensemble was chosen to simulate the system with molecular dynamics at 298 K at one atmospheric pressure. The time step was 1 fs, and the total simulation time was 300 ps. After balancing, the average density of the system was calculated. Then the Particle number, Volume, Temperature(NVT) ensemble was selected to simulate the dynamics of the system. The temperature was 298 K, the time step was 1fs, and the total simulation time was 200 ps. One frame was output every 2 ps, a total of 100 frames. The COMPASS force field was used to optimize the system in all dynamic simulations, which were completed through the Fortite module. The charge group method was used for the interaction between van der Waals and Coulomb, and the truncation radius was 9.5 nm. The charge of each anion and cation was assigned by using the current method, and the universal force field was applied to define the potential energy [33].

## 3. Experimental Results and Analysis

### 3.1. Microstructure

According to ASTM A923-2014, the polished sample of 2205 was etched in 40 g reagent grade NaOH plus 100 g water solution weight at 2 V DC for 20 s to obtain the structure in Figure 1. It can be seen that the microstructure of 2205 duplex stainless steel consists of an elongated austenite phase (γ, light color) inlaid on a ferrite matrix (α, Dark), the two phases are evenly distributed, and the contents of ferrite and austenite phases are 49.13% and 50.87% respectively.

### 3.2. Optimization Calculation and Design of Corrosion Inhibitor

The quantum chemical method was used to calculate the geometric optimization of the corrosion inhibitor, and the corresponding frontier orbital charge distribution was obtained. The results are shown in Figure 2. The electronic density of the nucleophilic frontier orbital (HOMO) of the organic matter in the corrosion inhibitor is mainly distributed on the O and N atoms in the molecule (Figure 2a), and the electronic density of the electrophilic frontier orbital (LUMO) is respectively distributed on the benzene ring (Figure 2b, providing electrons or receiving electrons on the 4 s orbital of the Fe atom respectively.

The adsorption behavior of the corrosion inhibitor and Fe surface in an aqueous solution was simulated by molecular dynamics, and the kinetic equilibrium configuration of the corresponding system at different temperatures was obtained (Figure 3). Figure 4 shows the adsorption mechanism of organic corrosion inhibitors on the iron surface. The combination of chemical adsorption and physical adsorption could form a protective film to reduce the corrosion of acid solution on the metal surface. When the corrosion inhibitor interacts with the iron surface, the O and N atoms in the benzene ring and molecule are close to the iron surface and tend to adsorb on the iron surface in parallel to form an adsorption layer. The alkyl chain deviates from the iron surface at a certain angle to form a thicker hydrophobic layer (Figure 3 and Figure 4). With the increase in temperature, the planarity of adsorption between the benzene ring and the iron surface becomes weaker, indicating that the adsorption capacity of the molecule and iron surface would decrease with the increase in temperature (Figure 3c).

The interaction energy between the inhibitor molecule and the iron surface at different temperatures was calculated according to the track file collected during the dynamic simulation [37].
(2)$$\Delta E=E_{inhibitor+Fe}-E_{Fe}-E_{inhibitor}$$
where Δ*E* is the interaction energy of the corrosion inhibitor and iron surface, *E_inhibitor+Fe_* is the total energy of the corrosion inhibitor and iron surface, *E_Fe_* is the energy of the iron surface, and *E_inhibitor_* is the energy of the inhibitor molecule. The binding energy is defined as the negative value of the interaction energy, i.e., *E_binding_* = −Δ*E*.

Table 2 depicts the calculated binding energy of the corrosion inhibitor and iron surface in an aqueous solution at different temperatures. The binding energy of the corrosion inhibitor and iron surface decreases gradually with the increase in temperature. The temperature has a great impact on the adsorption performance of corrosion inhibitors. Desorption is easy to occur with the increase in temperature, thus reducing its adsorption performance.

Corrosion inhibitor molecules could adsorb on the metal surface to form a protective film exhibiting corrosion inhibition performance, which slows down the anode reaction and the diffusion of metal ions, thus preventing the corrosion of the corrosive medium particles on the metal. The most important criterion of corrosion inhibition performance is whether corrosion inhibitors can adsorb on the metal surface to form a stable, protective film and maintain a high coverage for a long period of time, and can effectively prevent the migration of corrosive medium particles to the metal surface so as to block the reaction path of corrosion, which could be judged by the diffusion of the corrosion medium particles in the corrosion inhibitor film: “the stronger the diffusion ability, the worse the barrier performance of the film and the corrosion inhibitor molecules, on the contrary, the corrosion inhibition is stronger [38].

The diffusion behavior of corrosion medium particles H_2_O, H_3_O^+^, and Cl^−^ in corrosion inhibitor film was studied by molecular dynamics simulation. The diffusion coefficient of corrosion medium particles in the inhibitor film and the interaction energy between the inhibitor film and two kinds of corrosion medium particles were calculated. The diffusion coefficient is the most direct measure of the diffusion and migration ability of particles in the system. The larger the diffusion coefficient, the stronger the diffusion and migration ability of particles in the system, and the weaker the diffusion and migration ability of particles in the system.

According to Einstein’s relation [39], the diffusion coefficient can be expressed as Equations (3) and (4):(3)$$D=\frac{1}{6}\underset{t\to \infty }{\lim}\frac{d}{dt}\overset{n}{\underset{i}{\sum }}{\left\langle \left|R_{i}\left(t\right)-R_{i}\left(0\right)\right|\right\rangle }^{2}$$
(4)$$D=\frac{1}{6}\underset{t\to \infty }{\lim}\frac{dMSD}{dt}$$

By applying the finite difference approximation, the diffusion coefficient can be expressed as Equation (5):D = *m*/6(5)
where *m* is the slope of the MSD curve, and its value can be directly calculated by the software.

Table 3 shows the diffusion coefficients of H_2_O and corrosive medium particles in the inhibitor film. It shows that the diffusion coefficient is significantly reduced compared with that in the water system. The corrosion inhibitor film has a good blocking effect on the diffusion of corrosive medium particles, which could avoid its migration to the metal interface as much as possible and effectively inhibit the corrosion of metals.

Table 4 shows the diffusion coefficients of Cl^−^ in inhibitor film at different temperatures. The diffusion coefficient of Cl^−^ is relatively large in an aqueous solution, and it is easy to diffuse and migrate, resulting in corrosion pits. After the addition of the corrosion inhibitor, the diffusion coefficient decreases sharply, indicating that the diffusion and migration of Cl^−^ were effectively prevented and pitting is mitigated. The diffusion coefficient increases with the increase in temperature and the pitting corrosion is accelerated.

The interaction will have a greater impact on the diffusion and migration behavior of particles in it. If the interaction energy is positive, the medium has a repulsive effect on the particle, which can accelerate the diffusion of the particle. On the contrary, which means the medium is attractive to hinder the diffusion of particles. The interaction energy between corrosion medium particles and corrosion inhibitor film can be expressed by the following Equation:*E = E*_film+particle_−*E*_film_−*E*_particle_(6)
where *E* represents the interaction energy of particle and film, *E*_film+particle_ is the total energy of particle and film, *E*_film_ is the energy of the corrosion inhibitor film, and *E*_particle_ is the energy of the corrosion medium particles.

Table 5 shows the interaction energy between corrosion medium particles and corrosion inhibitor film. It can be seen that the interaction energy of water molecules and the inhibitor film is much smaller than that of the interaction energy of H_3_O^+^ and the inhibitor film, indicating that there may be a stronger interaction between H_3_O^+^ and the inhibitor film, which has excellent corrosion inhibition effect.

Considering the serious desorption of organic corrosion inhibitors at high temperatures, the formulation is to combine organic matter and inorganic salts that can form complexes so as to achieve rapid and stable film formation at high temperatures.

### 3.3. Corrosion Inhibition Mechanism Analysis at High-Temperature and High-Pressure

#### 3.3.1. Effect of Temperature on Performance of Corrosion Inhibitor

Figure 5, Figure 6 and Figure 7 show the uniform corrosion rate, inhibitor film thickness, and local corrosion depth of 2205 duplex stainless steel in acid solution at 140 °C, 160 °C, and 180 °C, respectively. It can be seen that the uniform corrosion rate shows no significant difference between 140 °C and 160 °C but rises sharply at 180 °C, which is twice that at 160 °C, which is far lower than the commonly accepted 81 g·m^−2^·h^−1^ [41] and 26.8053 g·m^−2^·h^−1^ of the 2205 duplex stainless steel in the reported 120 °C 15 wt.% HCl + 1.5% wt. HF + 3 wt.% HAC + 5.1 wt.% acid corrosion inhibitor [12]. The thickness of both layers showed a decreasing trend with the increase in temperature. When combined with the local corrosion and the morphology of the corrosion inhibitor film, it can be seen that the density of the film decreases with the increase in temperature.

No obvious localized corrosion was observed on the specimen surface after the 140 °C test. But the sample surface appeared to have localized corrosion at 160 °C and 180 °C tests. The maximum localized corrosion depth of the 160 °C sample is 22 μm. The average localized corrosion depth is 10 μm. The depth of the corrosion pit is mainly concentrated in 5~10 μm. The maximum localized corrosion depth of 180 °C sample surface is 37 μm. The average localized corrosion depth is 16 μm. The depth of the corrosion pit is mainly concentrated at 11~15 μm. It can be seen that the localized corrosion depth increases with the increase in test temperature.

Figure 8 shows the SEM morphology of the corrosion inhibitor film at different temperatures. At 140 °C, the inner and outer films of the corrosion inhibitor on the surface of the sample were relatively intact without obvious damage. With the increase in temperature, the corrosion inhibitor film was damaged, and the inner and outer films on the surface of the 180 °C sample were damaged. Thus, localized corrosion occurred.

The EDS results of the main element contents of outer and inner films are displayed in Table 5. The main elements of the film are C, O, Sb, Cl, Fe, Sand Cr, and the content of C in the outer film is much higher than that of in the inner film, while the content of Sb in the inner film is much higher than that of in the outer film. The contents of C and Sb indicate that the outer layer is mainly organic film, and the inner layer is mainly inorganic salt. The content of Cl^−^ in the outer film is higher than that of in the inner film, indicating that the corrosion inhibitor film has an excellent blocking effect in the acid solution, which is consistent with the simulation results (see Table 6). The content of Fe in the inner film increases with the increase in temperature, indicating that the corrosion resistance of the film is reduced.

#### 3.3.2. Effect of Time on Performance of Corrosion Inhibitor

The uniform corrosion rate of 2205 duplex stainless steel after 4 h and 12 h tests in acid solution at 180 °C is shown in Figure 9. It can be seen with the extension of test cycles, and the corrosion rate generally shows a decreasing trend. Figure 10 shows the thickness of two corrosion inhibitor film layers at different test cycles. With the extension of the test cycles, the thickness of the two layers displays an increasing trend. According to the local corrosion situation and the micro-morphology of the inhibitor film, although the film thickness increases with the increase in temperature, the compactness decreases, and the localized corrosion density and depth increase.

The cross-section and surface micromorphology of the inhibitor film at different test cycles shows in Figure 11. The two film layers are relatively complete in the fresh acid medium at 180 °C, which has a good protective effect on the substrate, but the local film layer is damaged, which leads to pitting corrosion on the surface of the sample. Although the thickness of the film in the 12 h test was significantly larger than that in the 4 h test, the compactness of the film was not significantly improved, and the degree of damage was more serious than that in the 4 h test.

Figure 12 and Figure 13 display the statistical results of pitting depth and the surface microscopic corrosion morphology of samples after different test cycles. It can be seen from Figure 12a that the maximum pitting depth is 37 μm, the average pitting depth is 16 μm, and the depth of the corrosion pit is mainly concentrated at 11~15 μm at 180 °C for 4 h. The maximum pitting depth is 64 μm, the average pitting depth is 23 μm, and the depth of the corrosion pit is mainly concentrated in 21~25 μm at 180 °C for 12 h (see Figure 12b).

#### 3.3.3. Corrosion Inhibitor Layer Analysis

In the acid solution system with high temperature and high concentration, the common organic corrosion inhibitor is seriously desorbed at high temperature, and its adsorption capacity differs greatly from that of austenite and ferrite, which is prone to selective corrosion and difficult to play a good protective role. The corrosion inhibitor in this study constructs a double-layer membrane adsorption structure, which uses inorganic substances with good stability at high temperature to form a complex with the metal matrix to enhance the binding ability with the metal matrix and maintain the high-temperature stability of the film. However, the inorganic film is not dense enough; organic film is used to supplement, as shown in Figure 14a. The microscopic morphology of the film cross-section after adding corrosion inhibitor in the 180 °C acid solution is displayed in Figure 14b. It can be seen the inhibitor has a double-layer membrane structure, which is consistent with the expected results of the inhibitor mechanism. Inorganic salt has a good binding ability with metal matrix at high-temperature. However, the film is easy to crack at high-temperature and forms a channel for acid diffusion. The crack defect could be repaired by the organic film, making the corrosion inhibitor film more dense and better protective for the matrix.

The cross-section of the line scan results of the corrosion inhibitor film is shown in Figure 15. The outer film and the inner film is mainly composed of element C and Sb, respectively, which is consistent with Table 5. From the results of line scanning, it can be seen that the organic film and the inorganic salt film penetrate into each other and combine closely.

Figure 16 shows the cross-section morphology of 2205 duplex stainless steel after testing in 180 °C acid solution with a corrosion inhibitor. No selective corrosion was observed. Figure 17 shows the XRD analysis results of the sample surface after the 180 °C test without corrosion inhibitor and with corrosion inhibitor. The peaks of the austenitic phase are much stronger than that of the ferritic phase in tests without the corrosion inhibitors. However, the peaks of the ferritic phase are as strong as those of the austenitic phase when the corrosion inhibitor is added. The results show that without adding a corrosion inhibitor, selective ferrite corrosion could be observed in high-temperature and high-concentration acid solutions, and selective corrosion is effectively inhibited after adding a corrosion inhibitor.

## 4. Discussion

Figure 18 shows the corrosion mechanism of duplex steel in a high-temperature and high-concentration acid solution system. Minor differences could be seen in the element composition of 2205 duplex stainless steel. Cr, Mo, Ni, and N tend to be concentrated in the ferrite and austenite phases, respectively [26,42,43]. The distribution of elements in the phase is a key factor affecting selective corrosion. The 2205 duplex stainless steel is in an active state when working with high-temperature and high-concentration hydrochloric acid and hydrochloric acid plus hydrofluoric acid [12,26]. Because the potential of Cr is more negative than that of Fe, Cr is more likely to lose electrons and form ions than Fe, and the corrosion of ferrite is more serious than that of austenite. The possible chemical reactions are:

anodic [44]:(7)$$\mathrm{Cr}\to {\mathrm{Cr}}^{3+}+3e$$
(8)$$\mathrm{Fe}\to {\mathrm{Fe}}^{2+}+2e$$
cathodic [12]:(9)$$2H^{+}+e\to H_{2}$$

Figure 19 shows the corrosion inhibition mechanism of acid corrosion inhibitors in high-temperature and high-concentration acid solution systems. The passive film of 2205 duplex stainless steel was dissolved in those harsh environments. Using high-temperature acid corrosion inhibitors can promote the steel to form a protective film on its surface, thus isolating the metal from the acid solution system and reducing the corrosion process. The SEM (see Figure 14) and EDS (see Figure 15) results all confirmed that the inhibitor film shows a double layer, and the inorganic film combined with the substrate forms a coordination bond, which enhances the stability of the inhibitor film at high-temperature. The outer layer is an organic layer. The quantum chemical calculation results show that the organic corrosion inhibitor molecule mainly takes the benzene ring and the O and N atoms in the molecule as the main adsorption sites. The O and N atoms have solitary electron pairs, which will coordinate with the empty orbit on the metal surface in the anode region to form an adsorption film [45] or adsorb at the defects of the inorganic film layer so as to improve the overall compactness of the film layer. The results of molecular dynamics simulation of the diffusion behavior of corrosion medium particles in the organic corrosion inhibitor film also show a good blocking effect on the corrosion media and improves the corrosion resistance of the double-layer film. The simulation result is confirmed by the EDS test results in Table 5, the content of Cl^−^ in the inner membrane is far lower than that of in the outer membrane, and the outer film provides a good barrier and supplement for the inner membrane. When the temperature rises from 140 °C~180 °C, the binding energy of the organic corrosion inhibitor decreases from 188.39 kcal·mol^−1^ to 180.75 kcal·mol^−1^. The high-temperature corrosion simulation test results demonstrate that there is no obvious local corrosion on the surface of the sample at 140 °C, and there is obvious local corrosion on the surface of the test sample at 160–180 °C, and the pitting corrosion is deeper at higher temperatures.

The corrosion inhibitor bilayer film is relatively dense when the temperature is low (see Figure 19a). The inorganic film is cracked or damaged with the increase in the experimental temperature, and the organic matter is adsorbed, which plays a supplementary role in the film layer (see Figure 19b). However, the binding energy of the organic matter and the adsorption property is reduced with the increase in temperature. Parts of the damaged film could not be repaired, leading to the base metal contact with the acid solution, then localized corrosion occurred (see Figure 19c). Although a thicker film could be formed with the further extension of test cycles, a loose inorganic film would be formed in the inner layer, and the organic film on the outer layer was also not intact. Hence, the local corrosion depth and scope were greatly increased (see Figure 19c). To sum up, the compactness of corrosion inhibitor film will be affected by the higher testing temperature and longer cycles.

## 5. Conclusions

In this paper, the corrosion inhibition performance of the self-designed ultra-high temperature acidizing corrosion inhibitor was studied by using quantum chemical calculation, MD simulation, SEM, EDS, and XRD. The main conclusions are summarized as follows:(1)The 2205 duplex stainless steel ultra-high-temperature acidizing corrosion inhibitor showed excellent corrosion effect in the 180 °C 9 wt. % HCl + 1.5 wt. % of HF + 3 wt. % CH_3_COOH acid system, the corrosion rate can be reduced to 12.1881 g·m^−2^·h^−1^ and is far below the commonly accepted 81 g·m^−2^·h^−1^, and no selective corrosion was observed.(2)The ultra-high temperature acidizing corrosion inhibitor has a double-layer membrane structure, and the inorganic membrane bonded with the substrate is mainly composed of Sb element, and the outer layer is an organic membrane mainly composed of C element.(3)The ultra-high temperature acidizing corrosion inhibitor had a good blocking effect on the diffusion of corrosive medium particles. With the increase in temperature, the binding energy of the corrosion inhibitor and substrate decreases, the blocking effect of corrosion inhibitor film on Cl^−^ also decreases, and pitting corrosion could easily be detected.(4)With the increase in temperature and the extension of test time, the density of corrosion inhibitor film could be damaged, and the corrosion inhibition effect decreased, which was manifested as the increase in local corrosion depth and diameter.(5)The developed acidizing corrosion inhibitor has great engineering application value to broaden the selection of materials for ultra-high-temperature and high-pressure wells and reduce the cost of oil and gas field string.

## Figures and Tables

**Figure 1 materials-16-02358-f001:**
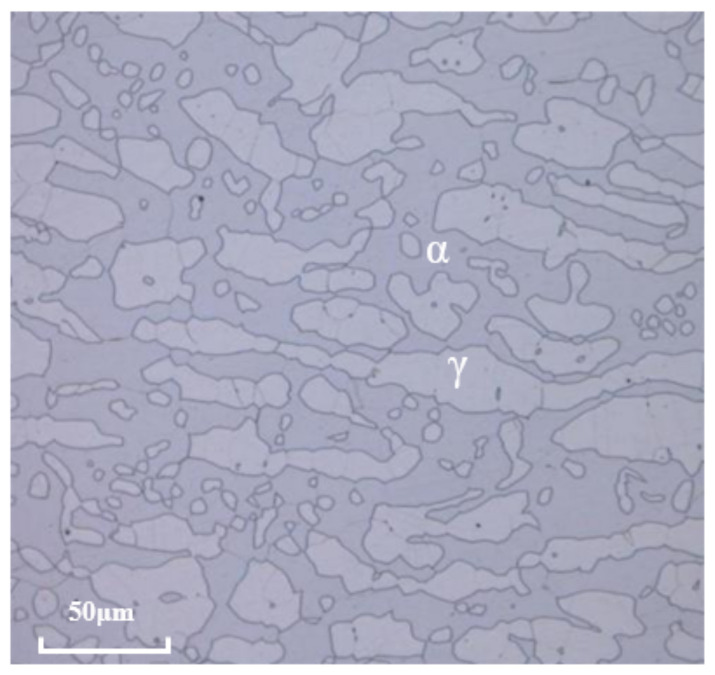
Metallographic morphology of 2205 duplex stainless steel for test.

**Figure 2 materials-16-02358-f002:**
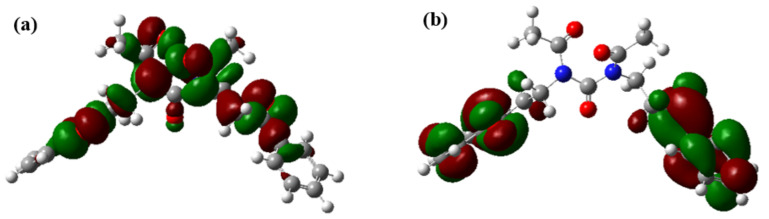
Charge distribution of organic matter frontier orbital in corrosion inhibitor ((**a**)-nucleophilic frontier orbital HOMO, (**b**)-electrophilic frontier orbital LUMO).

**Figure 3 materials-16-02358-f003:**
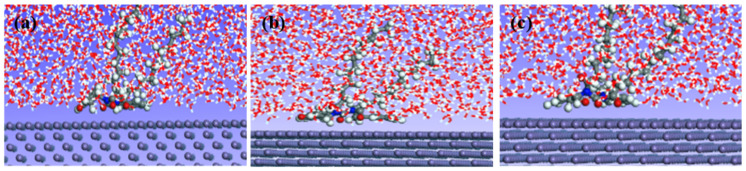
Adsorption behavior of corrosion inhibitor and Fe surface in different temperature systems. ((**a**) 140 °C, (**b**) 160 °C, (**c**) 180 °C).

**Figure 4 materials-16-02358-f004:**
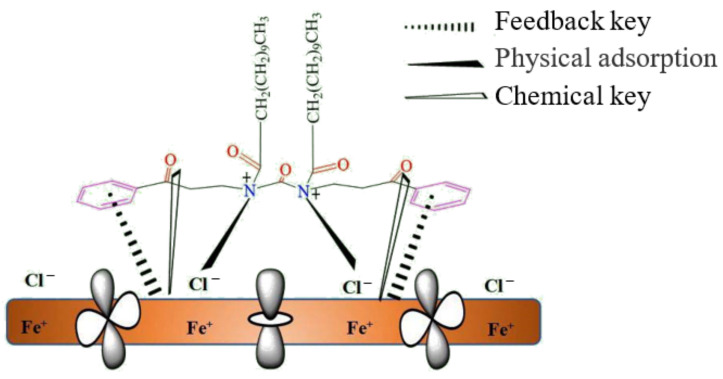
Adsorption mechanism of organic corrosion inhibitor on iron surface.

**Figure 5 materials-16-02358-f005:**
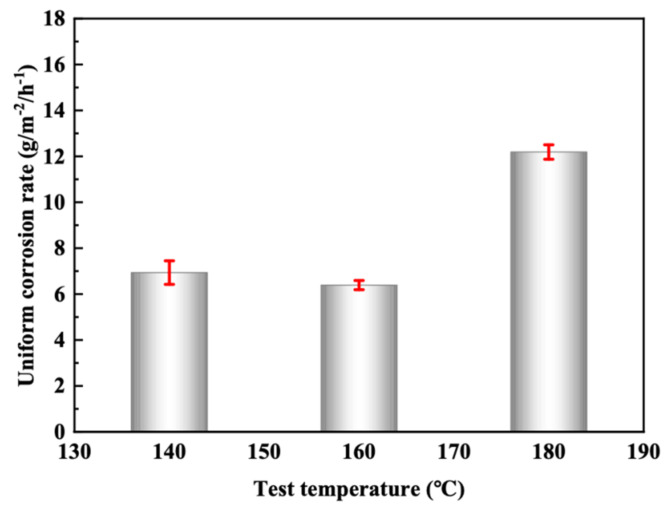
Uniform corrosion rate of 2205 duplex stainless steel at different temperatures.

**Figure 6 materials-16-02358-f006:**
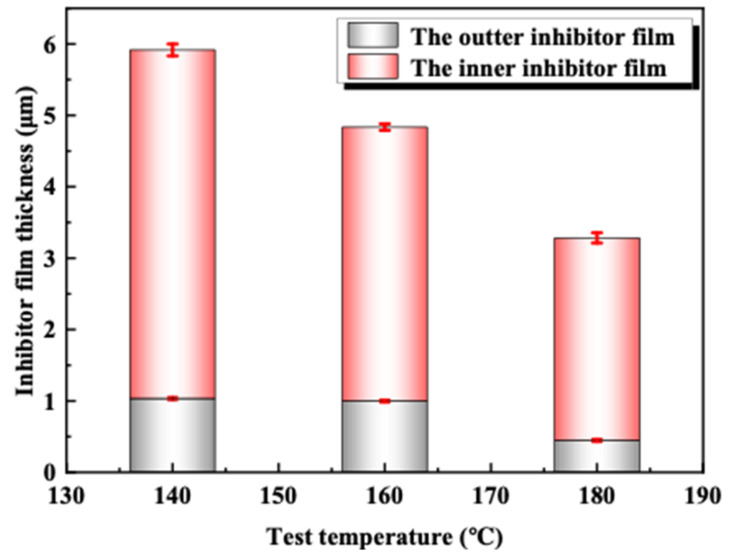
Thickness of corrosion inhibitor film on 2205 duplex stainless steel at different temperatures.

**Figure 7 materials-16-02358-f007:**
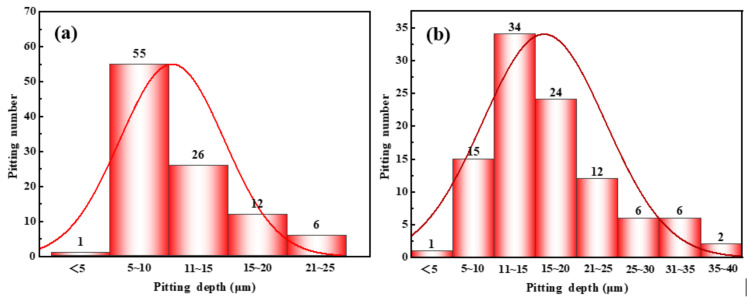
Localized corrosion depth of 2205 duplex stainless steel in acid solution system at different temperatures ((**a**) 160 °C; (**b**) 180 °C).

**Figure 8 materials-16-02358-f008:**
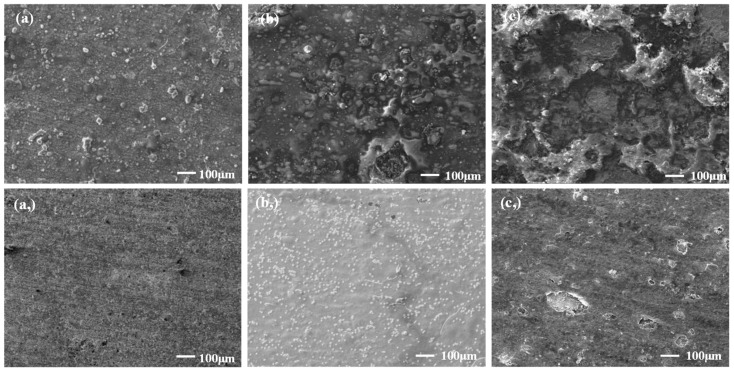
Micromorphology of corrosion inhibitor films on the surface of 2205 duplex stainless steel sample at 100× magnification ((**a**) 140 °C the outer film, (**a,**) 140 °C the inner film, (**b**) 160 °C outer film, (**b,**) 160 °C the inner film, (**c**) 180 °C outer film, (**c,**) 180 °C the inner film).

**Figure 9 materials-16-02358-f009:**
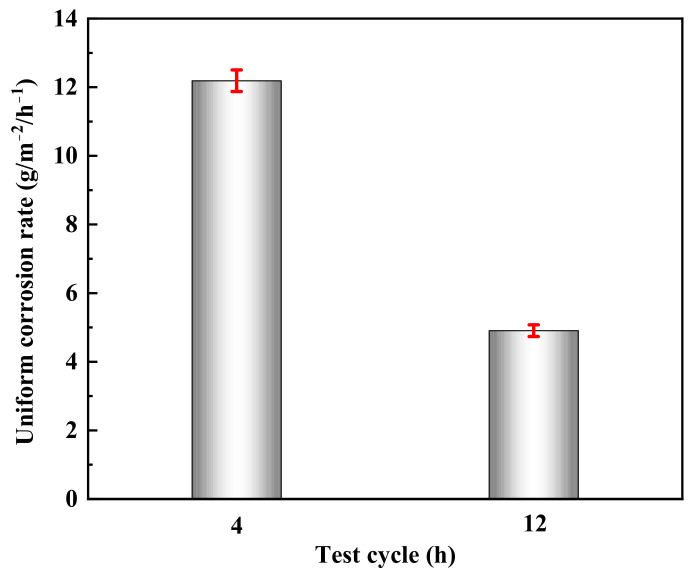
Uniform corrosion rate at different test cycle.

**Figure 10 materials-16-02358-f010:**
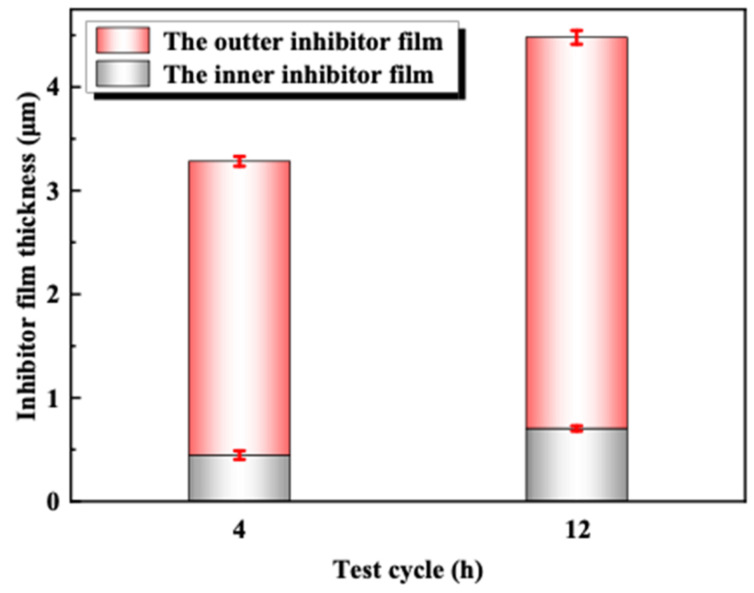
Thickness measurement results of corrosion inhibitor film at different test cycles.

**Figure 11 materials-16-02358-f011:**
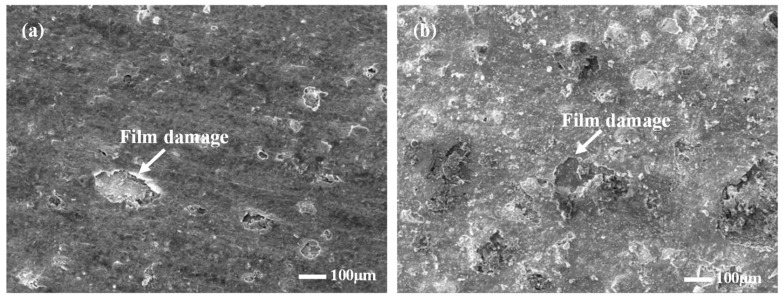
Micromorphology of the inner film of the corrosion inhibitor on the surface of the samples at different test cycles magnified by 100× ((**a**) test cycle 4 h; (**b**) test cycle 12 h).

**Figure 12 materials-16-02358-f012:**
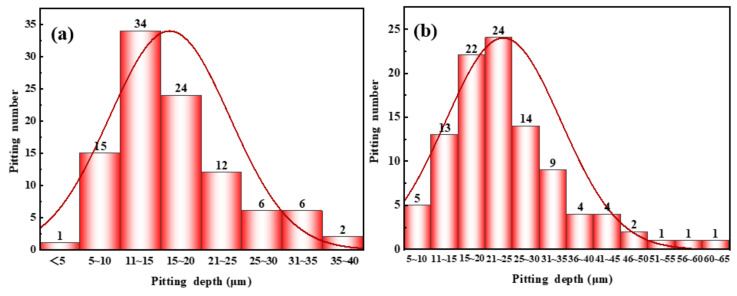
Normal distribution of pitting depth on the surface of samples at different test cycles (**a**) 4 h; (**b**) 12 h.

**Figure 13 materials-16-02358-f013:**
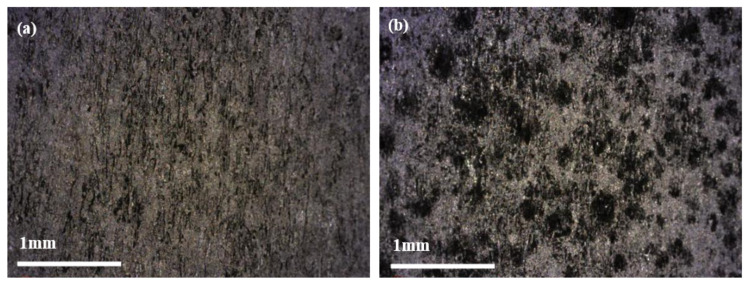
Micromorphology of sample surface magnified by 100× under optical microscope at different test cycles: (**a**) 4 h, (**b**) 12 h.

**Figure 14 materials-16-02358-f014:**
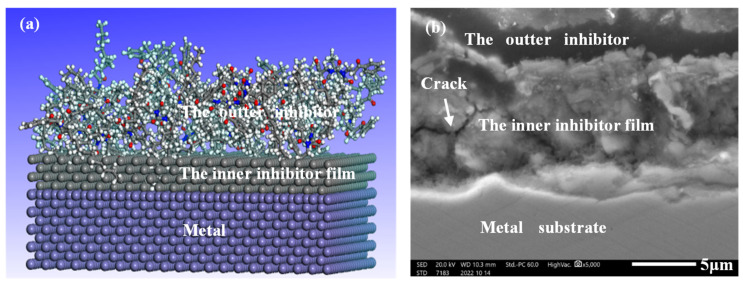
Structure of high-temperature acidizing corrosion inhibitor double-layer film ((**a**) schematic diagram, (**b**) section SEM).

**Figure 15 materials-16-02358-f015:**
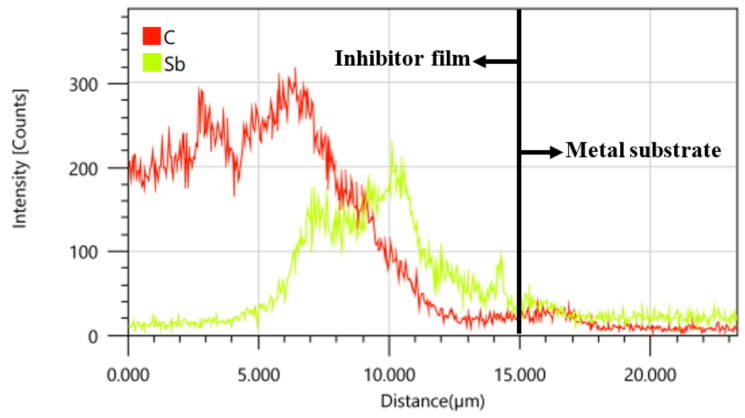
Line Scanning Results of Element Distribution of Corrosion Inhibitor Film.

**Figure 16 materials-16-02358-f016:**
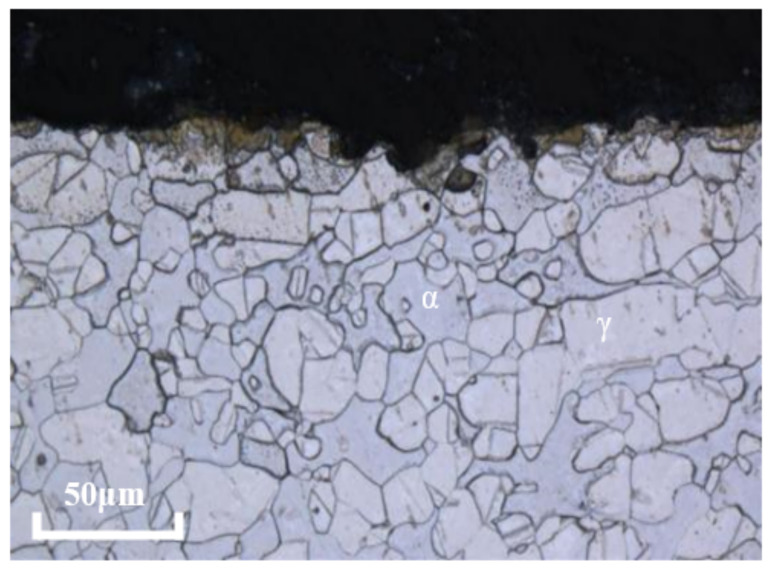
Cross-sectional morphology of 2205 duplex stainless steel sample without corrosion product film after testing in 180 °C acid solution with corrosion inhibitor.

**Figure 17 materials-16-02358-f017:**
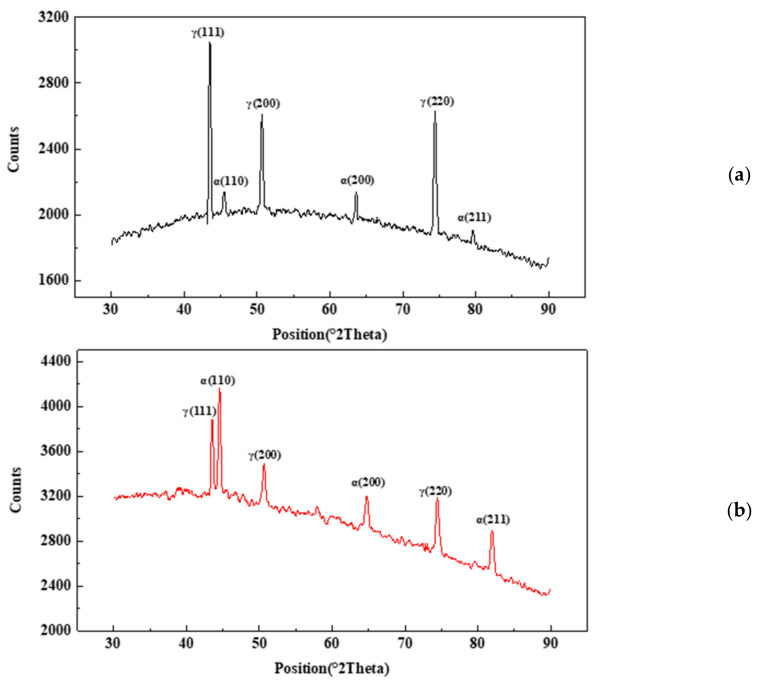
XRD analysis results of samples without and with corrosion inhibitor ((**a**) 180 °C without inhibitor, (**b**) 180 °C with inhibitor).

**Figure 18 materials-16-02358-f018:**
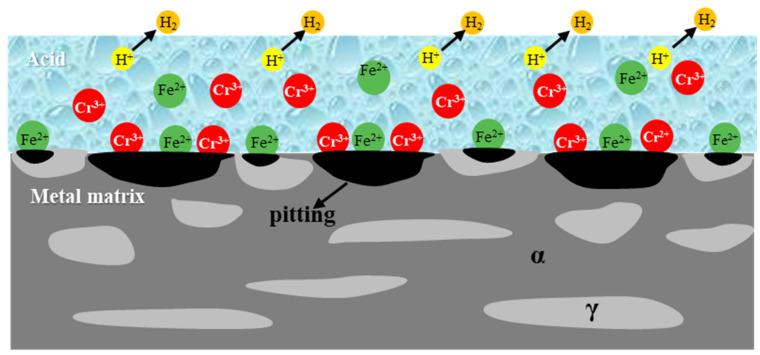
Corrosion mechanism of duplex stainless steel in high-temperature and high-concentration acid solution system.

**Figure 19 materials-16-02358-f019:**
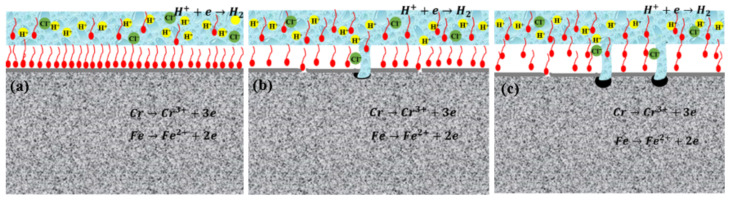
Schematic diagram of corrosion inhibition mechanism of acidizing corrosion inhibitor for duplex stainless steel in high-temperature and high-concentration acid solution system ((**a**) 140 °C, 4h (**b**) 180 °C, 4 h, (**c**) 180 °C, 12 h).

**Table 1 materials-16-02358-t001:** The specific corrosion inhibitor test conditions of 2205 duplex stainless steel in ultra-high-temperature acidizing.

Medium	Temperature (°C)	Time (h)	Concentration (wt.%)
9 wt.%HCl + 1.5 wt.% HF + 3 wt.% CH_3_COOH	140	4	4
	160	4	5
	180	4	6
	180	12	6

**Table 2 materials-16-02358-t002:** Binding energy of corrosion inhibitor and iron surface in aqueous solution at different temperatures.

Temperatures (°C)	140	160	180
Binding energy (kcal/mol)	188.39	183.26	180.75

**Table 3 materials-16-02358-t003:** Diffusion coefficient of two corrosion media particles in the inhibitor film, the self-diffusion coefficient of water is 2.34 × 10^−9^ m^2^/s.

System	Corrosion Inhibitor -H_2_O	Corrosion Inhibitor -H_3_O^+^
Diffusion coefficient (10^−9^ m^2^/s)	0.0108	0.0099

**Table 4 materials-16-02358-t004:** Diffusion coefficients of Cl^−^ in corrosion inhibitor films at different temperatures.

System	H_2_O [40]	Corrosion Inhibitor		
Temperature		140	160	180
Diffusion coefficient (10^−9^ m^2^/s)	0.3220	0.0177	0.0189	0.0226

**Table 5 materials-16-02358-t005:** Interaction energy of corrosion medium particles and corrosion inhibitor film.

System	Corrosion Inhibitor Film -H_2_O	Corrosion Inhibitor Film -H_3_O^+^
*E*(kcal/mol)	−99.25	−155.91

**Table 6 materials-16-02358-t006:** Main element contents EDS analysis result of outer and inner films at different test temperatures (wt.%).

Temperature (°C)	Test Location	C	O	F	S	Cl	Cr	Fe	Ni	Mo	Sb
140	Outer film	40.38	7.39	-	1.27	5.42	0.37	2.22	0.52	-	42.43
	Inner film	4.63	2.94		0.48	0.89	1.35	5.3	-	-	78.99
160	Outer film	53.95	7.47	1.84	0.98	7.77	-	1.23	-	-	26.33
	Inner film	3.12	-	-	-	-	1.26	5.30	-	-	89.64
180	Outer film	64.55	7.62	-	1.04	7.81	0.52	1.82	2.36	-	14.28
	Inner film	6.84	7.64	-	0.28	1.84	5.54	15.22	18.32	1.78	42.53

## Data Availability

Not applicable.

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
