# Peer review of "Corrosion Inhibition Mechanism of Ultra-High-Temperature Acidizing Corrosion Inhibitor for 2205 Duplex Stainless Steel"

_materials, 2023, doi:10.3390/ma16062358_

Round 1

Reviewer 1 Report

The main weakness of the article are :

1-Please rewrite the first paragraph of your abstract, make it consist of some coherence sentences.

2-three value for corrosion resistance, they belong to what condition?

3-The english is very poor and should be improved.

4-please shorten the paragraph 2 and 3 of your introduction, be more focus about the recent progress in other studies

5-There is no literature review in introduction about the relation between pitting corrosion and Md simulation,

To fix this problem the following study should be added to the introduction and prove there is a straight relation between MD simulation and pitting corrosion:

https://doi.org/10.1016/j.diamond.2023.109793

6- in introduction add unit every time the number or precentage is mentionef

7- The unit of duplex stainless steel chemical composition is missing in experimental section

8-what is the concentration in table 1?

9- There is missing information about potential and particle charge in MD simulation, use the following article and input all the information based on this article

https://doi.org/10.1016/j.diamond.2023.109793

10- The EDS is missing is SEM results, it should be added

11-The whole conclusion should be rewrite

Author Response

Thank you so much for your time spending on reviewing this paper. Any comments from you are greatly appreciated by the authors. Some comments are very helpful to our future work.

The modification in the revised paper to the comments:

Point 1: Please rewrite the first paragraph of your abstract, make it consist of some coherence sentences.

Response 1

We have rewitten the abstract.

Point 2: three value for corrosion resistance, they belong to what condition

Response 2

Thanks for your comment. The test conditions have been supplemented.

Point 3: The english is very poor and should be improved.

Thank you very much for your kind advice. The English of the whole manuscript has been improved.

Point 4: please shorten the paragraph 2 and 3 of your introduction, be more focus about the recent progress in other studies

Response 4

It has been revised according to the requirements.

Point 5:There is no literature review in introduction about the relation between pitting corrosion and Md simulation, To fix this problem the following study should be added to the introduction and prove there is a straight relation between MD simulation and pitting corrosion https://doi.org/10.1016/j.diamond.2023.109793

Response 5

Thanks for your comment. Relevant literature has been added to the Introduction part. And the recommended paper on MD simulation about Cl- and corrosion inhibitor has been added.

Point 6: in introduction add unit every time the number or precentage is mentionef

Response 6

It has been revised according to the requirements.

Point 7:The unit of duplex stainless steel chemical composition is missing in experimental section

Response 7

It has been revised according to the requirements.

Point 8:what is the concentration in table 1?

Response 8

The concentration in Table 1 is added.

Point 9:There is missing information about potential and particle charge in MD simulation, use the following article and input all the information based on this article

https://doi.org/10.1016/j.diamond.2023.109793

Response 9

Thanks for your comment. The missing information has been added according to above paper.

Point 10: The EDS is missing is SEM results, it should be added

Response 10

The EDS result is added.

Point 11: The whole conclusion should be rewrite

Response 11

We have rewritten the conclusion.

Reviewer 2 Report

Dear colleagues,

  Studies on industrial applications are quite remarkable.  I enjoyed reading your article. The article is well planned and written. Reviewing the following points will further strengthen the article. Therefore, a minor revision is required.

1-      After Line 50, Stainless steels, their types, their differences and importance should be mentioned as a paragraph. It should be noted with a paragraph that duplex stainless steels contain the positive aspects of the properties of austenitic and ferritic stainless steels. The following references can be used for stainles steel, austeninitic and ferritic stainless steels.

https://doi.org/10.1016/j.mser.2009.03.001

https://doi.org/10.1134/S2070205114010195

https://doi.org/10.1016/j.corsci.2010.01.037

https://doi.org/10.1108/ACMM-12-2012-1224

2-      In the continuation of Figure 1, the etching used inshould be given.

3-      If the data in Figure 5 and Figure 6 were not obtained upon examination of a single sample, standard deviations should be shown on the figures. At least 3 test averages are required for experiments. The same is true for most charts.

4-      If possible, the contents of the corrosion products on Figure 8 can be determined by EDS or XPS analysis.

5-      The XRD peaks given in Figure 17 must be defined. Which peaks occur at which temperatures should be examined in detail.

6-      The industrial achievements of the study can be added to the Conclusion as a point.

Author Response

Thank you so much for your time spending on reviewing this paper. Any comments from you are greatly appreciated by the authors. Some comments are very helpful to our future work.

The modification in the revised paper to the comments:

Point 1 After Line 50, Stainless steels, their types, their differences and importance should be mentioned as a paragraph. It should be noted with a paragraph that duplex stainless steels contain the positive aspects of the properties of austenitic and ferritic stainless steels. The following references can be used for stainles steel, austeninitic and ferritic stainless steels.

https://doi.org/10.1016/j.mser.2009.03.001

https://doi.org/10.1134/S2070205114010195

https://doi.org/10.1016/j.corsci.2010.01.037

https://doi.org/10.1108/ACMM-12-2012-1224

Response 1

Thanks for your comment. It has been revised according to the comment.

Point 2 In the continuation of Figure 1, the etching used in should be given

Response 2

We have given the etching method of Fig. 1.

Point 3 If the data in Figure 5 and Figure 6 were not obtained upon examination of a single sample, standard deviations should be shown on the figures. At least 3 test averages are required for experiments. The same is true for most charts.

Response 3

It has been revised according to the comment.

Point 4 If possible, the contents of the corrosion products on Figure 8 can be determined by EDS or XPS analysis.

Response 4

Thanks for your comment.The EDS result has been added.

Point 5 The XRD peaks given in Figure 17 must be defined. Which peaks occur at which temperatures should be examined in detail.

Response 5

We have improved on these deficiencies.

Point 6 The industrial achievements of the study can be added to the Conclusion as a point.

Response 6

Thanks for your advice. We have rewritten the Conclusion part.

Round 2

Reviewer 1 Report

The article can be published at this current form.